# A Randomized Trial on Resveratrol Supplement Affecting Lipid Profile and Other Metabolic Markers in Subjects with Dyslipidemia

**DOI:** 10.3390/nu15030492

**Published:** 2023-01-17

**Authors:** Yuqing Zhou, Yupeng Zeng, Zhijun Pan, Yufeng Jin, Qing Li, Juan Pang, Xin Wang, Yu Chen, Yan Yang, Wenhua Ling

**Affiliations:** 1Department of Nutrition, School of Public Health, Sun Yat-sen University, Guangzhou 510080, China; 2Guangdong Provincial Key Laboratory of Food, Nutrition and Health, Guangzhou 510080, China; 3Guangdong Engineering Technology Center of Nutrition Transformation, Guangzhou 510080, China; 4Department of Nutrition, School of Public Health (Shenzhen), Sun Yat-sen University, Shenzhen 518107, China

**Keywords:** resveratrol, uric acid, xanthine oxidase, dyslipidemia, randomized controlled trial

## Abstract

Resveratrol is a polyphenol with a well-established beneficial effect on dyslipidemia and hyperuricemia in preclinical experiments. Nonetheless, its efficacy and dose–response relationship in clinical trials remains unclear. This study examined whether resveratrol supplement improves the serum lipid profile and other metabolic markers in a dose-response manner in individuals with dyslipidemia. A total of 168 subjects were randomly assigned to placebo (*n* = 43) and resveratrol treatment groups of 100 mg/d (*n* = 41), 300 mg/d (*n* = 43), and 600 mg/d (*n* = 41). Anthropometric and biochemical parameters were analyzed at baseline and 4 and 8 weeks. Resveratrol supplementation for 8 weeks did not significantly change the lipid profile compared with the placebo. However, a significant decrease of serum uric acid was observed at 8 weeks in 300 mg/d (−23.60 ± 61.53 μmol/L, *p* < 0.05) and 600 mg/d resveratrol groups (−24.37 ± 64.24 μmol/L, *p* < 0.01) compared to placebo (8.19 ± 44.60 μmol/L). Furthermore, xanthine oxidase (XO) activity decreased significantly in the 600 mg/d resveratrol group (−0.09 ± 0.29 U/mL, *p* < 0.05) compared with placebo (0.03 ± 0.20 U/mL) after 8 weeks. The reduction of uric acid and XO activity exhibited a dose–response relationship (*p* for trend, <0.05). Furthermore, a marked correlation was found between the changes in uric acid and XO activity in the resveratrol groups (*r* = 0.254, *p* < 0.01). Resveratrol (10 μmol/L) treatment to HepG2 cells significantly reduced the uric acid levels and intracellular XO activity. Nevertheless, we failed to detect significant differences in glucose, insulin, or oxidative stress biomarkers between the resveratrol groups and placebo. In conclusion, resveratrol supplementation for 8 weeks had no significant effect on lipid profile but decreased uric acid in a dose-response manner, possibly due to XO inhibition in subjects with dyslipidemia. The trial was registered on ClinicalTrials.gov (NCT04886297).

## 1. Introduction

Dyslipidemia, which refers to elevated concentration of triglyceride, total cholesterol (TC), LDL cholesterol, and decreased concentration of HDL cholesterol, is a well-established risk factor for cardiovascular disease (CVD) [1]. Hyperuricemia, or high serum uric acid concentrations, is a common metabolic abnormality in patients with dyslipidemia [2]. It has been reported that 25.9% of individuals with dyslipidemia had hyperuricemia [3]. Epidemiological studies in different populations have demonstrated a close relationship between hyperuricemia and hyperlipidemia [4,5,6] and an independent association between increased serum uric acid concentration and elevated morbidity or mortality from CVD [7,8,9,10]. In addition, individuals with dyslipidemia often exhibit other metabolic abnormalities, such as oxidative stress, increased inflammatory factors, and insulin resistance [11,12].

Although the mechanism underlying the significant association between hyperuricemia and CVD is unclear, oxidative stress is widely thought to be one of the potential mechanisms [13,14]. Current evidence demonstrates that uric acid is mainly produced from xanthine catalyzed by xanthine oxidase (XO) in the liver and excreted through the kidneys and intestines [15]. As an endogenous antioxidant, uric acid accounts for the majority of the total antioxidant capacity of the human body [16]. However, when serum uric acid level is elevated, it is converted into a pro-oxidant to trigger the oxidative stress response, thereby contributing to the initiation and progression of atherosclerotic diseases [17,18,19,20].

Resveratrol is a polyphenolic compound mainly found in plants, including peanuts, grapes, knotweed, and so on [21,22]. Emerging evidence from in vivo and in vitro studies highlights the potential effects of resveratrol, including anti-oxidative stress [23], anti-inflammatory [24], anti-cancer [25], cardioprotection [26], renoprotection [27], and atherosclerosis regression [28]. Although the effect of resveratrol on the improvement of hyperlipidemia and hyperuricemia has been validated in animal studies [29,30,31], clinical trials have yielded inconsistent results [32,33], which may result from the heterogeneity in dose and duration of supplementation. Furthermore, it should be borne in mind that most intervention studies assessed the influence of a single dose of resveratrol on lipid profile or other metabolic markers. Therefore, we sought to demonstrate resveratrol’s efficacy and dose-response effect in subjects with dyslipidemia in this randomized controlled trial, with lipid profile as the primary outcome and uric acid, XO activity, glucose, insulin, and oxidative stress biomarkers as the secondary outcomes. Subsequently, we explored the mechanism by which resveratrol affects uric acid in the human hepatocellular carcinoma cell lines (HepG2 cells).

## 2. Materials and Methods

### 2.1. Subjects

All subjects were recruited at the Huanghuagang Street Community Health Service Center, Guangzhou, China. Recruitment methods included advertisements, on-site presentations, and recommendations by doctors. Participants were initially screened by face-to-face interviews and underwent blood biochemical studies to confirm their eligibility. The inclusion criteria were: (1) age 35–70 years; (2) dyslipidemia defined as two or more of these four criteria [34]: fasting serum triglyceride ≥ 1.70 mmol/L, TC ≥ 5.20 mmol/L, LDL cholesterol ≥ 3.12 mmol/L, or HDL cholesterol ≤ 0.91 mmol/L; and (3) willingness to maintain daily eating habits, physical activity patterns and body weight throughout the study. The exclusion criteria were: (1) intake of any medication affecting lipid metabolism, such as fibrates and statins, for the past 6 months; (2) use of any phytochemical supplementation, including resveratrol, for the past 3 months; (3) subjects with severe acute or chronic illness within the past 1 month; and (4) lactating or pregnant females. The trial was granted ethical approval by the ethical committee of the School of Public Health of Sun Yat-sen University and registered at ClinicalTrials.gov (NCT04886297).

### 2.2. Experimental Design

The study involved a randomized, double-blind, placebo-controlled design lasting eight weeks. A total of 168 subjects were randomly allocated to one of four treatment groups: placebo (*n* = 43), 100 mg/d resveratrol (*n* = 41), 300 mg/d resveratrol (*n* = 43), or 600 mg/d resveratrol (*n* = 41). The placebo and 100 and 300 mg resveratrol capsules (Mega Resveratrol, USA; refer to Appendix A for the certificate of analysis) had identical appearance and packaging. Subjects were asked to take two capsules with breakfast every morning for 8 weeks (the placebo group was prescribed two placebo capsules; the 100 mg/d group was allocated one placebo and one 100 mg resveratrol capsule; the 300 mg/d group was prescribed one placebo and one 300 mg resveratrol capsule; the 600 mg/d group was allocated two 300 mg resveratrol capsules). During the intervention period, subjects were followed up every two weeks to ensure compliance by counting the number of remaining capsules and to assess for adverse events. For dietary monitoring, subjects were interviewed by trained researchers to collect the 3-day 24 h dietary recalls at week 0 (baseline), 4 (mid-intervention), and 8 (end of intervention), and nutrient intake data were analyzed by the computer-aided nutritional analysis program (Chinese Food Composition Table).

### 2.3. Anthropometric Analyses

Anthropometric measurements were carried out by a trained researcher and included weight, height, BMI [weight (kg)/height^2^ (m^2^)], waist circumference (WC), hip circumference (HC), and waist-hip ratio (WHR) [WC (cm)/HC (cm)]. Systolic blood pressure (SBP), diastolic blood pressure (DBP), and heart rate (HR) were measured in a sitting position at rest using an electronic sphygmomanometer (maibobo, Shenzhen, China). All parameters measured were performed following standardized procedures and using regularly calibrated equipment. Information on sociodemographic characteristics and lifestyle habits was collected by trained researchers through interviews using structured questionnaires.

### 2.4. Determination of Biochemical Parameters

At weeks 0, 4, and 8, each subject was asked to fast for > 12 hours overnight, then blood and urine samples were collected the next morning. After centrifugation (at 3000 rpm, 4 °C for 15 min), aliquots of blood samples were stored at −80 °C until further analysis. Serum lipid profile (TC, LDL cholesterol, HDL cholesterol, triglyceride, apoA1, and apoB), uric acid, glucose, and superoxide dismutase (SOD) were analyzed by the Cobas c702 automatic biochemical analyzer (Roche Diagnostics, Basel, Switzerland). Serum XO activity was detected by commercial kits (Cat NO. BC1095, Solarbio, Beijing, China). Fasting insulin was measured by the Cobas e602 automatic biochemical analyzer (Roche Diagnostics, Switzerland). The HOMA-IR was calculated by the formula: HOMA-IR = [glucose (mmol/L) × insulin (mU/mL)]/22.5. Serum malonaldehyde (MDA, Cat NO. JL11466), allantoin (Cat NO. JL47385), glutathione S-transferase (GST, Cat NO. JL11241), glutathione (GSH, Cat NO. JL10703), glutathione peroxidase (GSH-Px, Cat NO. JL10355), urine 8-iso-prostaglandin F2α (8-iso-PGF2α, Cat NO. JL19022) and 8-hydroxy-2′-deoxyguanosine (8-OHdG, Cat NO. JL11850) were measured by ELISA kits (J&L Biological, Shanghai, China). Urine biomarker values were normalized to urine creatinine concentration, measured by the BS-830 automatic biochemical analyzer (Mindray, Shenzhen, China).

### 2.5. Cell Culture

HepG2 cells were obtained from Procell Life Science & Technology (Wuhan, China) and cultured in Dulbecco’s modified eagle medium with 10% fetal bovine serum, 100 U/mL penicillin, and 100 µg/mL streptomycin at 37 °C in 5% CO2. To establish an in vitro hyperlipidemia model, HepG2 cells were treated with free fatty acids (FFA), a mixture of oleate and palmitate with a final concentration of 750 µmol/L, and incubated with or without resveratrol (10 μmol/L) for 48 h. Uric acid levels in culture supernatants were measured by Mindray BS-830 automatic biochemical analyzer (Shenzhen, China). Intracellular XO activity was analyzed and normalized by the total protein measured by commercial kits (Cat NO. PC0020, Solarbio, Beijing, China). The protein expression of XO was analyzed by Western blot with primary antibodies against GAPDH (Cat NO. K200057M, Solarbio, Beijing, China) and XO (Cat NO. ab133268, Abcam, Cambridge, UK).

### 2.6. Sample Size Estimation

The sample size was estimated with the PASS software (version 15.0, NCSS Inc.). A previous trial [35] reported that 300 mg/d resveratrol led to a 0.41 mmol/L decrease in LDL cholesterol compared with the placebo. With a power of 90% and an alpha level of 0.05, 34 subjects were required for each group. Considering a dropout rate of 10%, a minimum of 38 subjects was required per group.

### 2.7. Statistical Analysis

All statistical analyses were conducted by SPSS software (version 25.0, SPSS Inc.). Categorical variables were analyzed by the chi-square test, while continuous variables were expressed as mean ± SD; one-way ANOVA was used to compare the means among four groups, followed by Bonferroni correction for post hoc comparisons if equal variances were assumed; otherwise, the Games–Howell test was applied. Comparisons between week 0 and weeks 4 or 8 within each group were performed using a paired Student’s *t*-test. The polynomial linear trend test was used to assess the dose–response relationship. The relationship between the two variables was assessed by Pearson correlation analysis. Differences in the uric acid levels, activity, and protein expression of XO in HepG2 cells were evaluated by the Student’s *t*-test for independent samples. *p* values <  0.05 were statistically significant.

## 3. Results

### 3.1. Tolerability and Compliance

Among the 187 participants, 19 were lost to follow-up in the placebo (*n* = 4), 100 mg/d (*n* = 5), 300 mg/d (*n* = 4), and 600 mg/d (*n* = 6) resveratrol groups. The placebo and resveratrol capsules were generally well tolerated, and no serious adverse reactions were reported during the trial. Most participants stopped taking the supplements for personal reasons, while the remaining few for mild adverse effects, with one person each in the placebo and 100 mg/d resveratrol groups reporting stomachache and one person in the 600 mg/d group reporting headache. The trial flow diagram is shown in Figure 1. Compliance with daily dosing of capsules exceeded 95% for all groups.

### 3.2. Baseline Characteristics and Dietary Monitoring

The baseline characteristics of subjects are summarized in Table 1. The four groups were comparable in age, sex, smoking status, physical activities, weight, BMI, WC, HC, WHR, SBP, DBP, HR, and other biochemical parameters, including the lipid profile, uric acid, glucose, insulin, HOMA-IR, and oxidative stress biomarkers (*p* > 0.05).

Table 2 shows the nutrient intake data at baseline and 8 weeks. No significant differences in energy, nutrient, or dietary resveratrol intake were found after 8 weeks of intervention (*p* > 0.05).

### 3.3. Effect of Resveratrol on Lipid Profile in Dyslipidemia

Analysis of changes in the lipid profile between baseline and follow-up in the four groups yielded no significant differences in triglyceride, TC, LDL cholesterol, HDL cholesterol, apoA1, apoB, or apoA1/apoB (Table 3).

### 3.4. Effect of Resveratrol on Uric Acid and XO Activity in Dyslipidemia

The absolute within-group changes in uric acid and XO activity from baseline to follow-up are presented in Table 4 and Figure 2. In the 300 mg/d resveratrol group, the concentration of uric acid was significantly decreased from 378.37 ± 102.62 μmol/L at baseline to 362.51 ± 93.51 μmol/L (*p* < 0.05) at 4 weeks and to 354.77 ± 89.00 μmol/L (*p* < 0.05) at 8 weeks. The 600 mg/d intervention group exhibited further reduction from 376.37 ± 95.94 μmol/L at baseline to 358.44 ± 87.31 μmol/L and 352.00 ± 87.52 μmol/L (*p* < 0.05) at 4 and 8 weeks, respectively. Resveratrol supplement with 600 mg/d decreased serum XO activity from 0.58 ± 0.32 U/mL to 0.50 ± 0.27 U/mL at 4 weeks and to 0.49 ± 0.26 U/mL at 8 weeks (although the difference was not statistically significant). However, 100 mg/d or 300 mg/d of resveratrol did not cause any significant reduction.

The relative changes in uric acid and XO activity among groups were compared and further analyzed (Table 4, Figure 3). The results showed that the uric acid levels at 8 weeks were significantly reduced in 300 mg/d resveratrol (−23.60 ± 61.53 μmol/L, *p* < 0.05) and 600 mg/d resveratrol group (−24.37 ± 64.24 μmol/L, *p* < 0.01) compared with the placebo (8.19 ± 44.60 μmol/L), which showed a dose-response trend (*p* for trend, <0.01). Moreover, XO activity in the 600 mg/d resveratrol group was significantly reduced compared to placebo at 8 weeks (−0.09 ± 0.29 U/mL vs. 0.03 ± 0.20 U/mL, *p* < 0.05). Although the change in XO activity was not significantly different among the four groups, a significant dose-dependent trend was found (*p* for trend, <0.05).

The correlation between the changes in uric acid and XO activity at 8 weeks is depicted in Figure 4. In the resveratrol groups, the change in uric acid concentration was positively correlated with XO activity (*r* = 0.254, *p* < 0.01), whereas no significant correlation was observed in the placebo.

### 3.5. Effect of Resveratrol on Other Parameters in Dyslipidemia

No significant changes were observed in glucose, insulin, HOMA-IR, SOD, MDA, allantoin, GST, GSH, GSH-Px, urine 8-iso-PGF2α, or 8-OHdG among groups (Table 5 and Table 6).

### 3.6. Effect of Resveratrol on Uric Acid and XO In Vitro

As presented in Figure 5, FFA treatment increased uric acid levels in the supernatants and intracellular XO activity of HepG2 cells, and resveratrol (10 μmol/L) significantly decreased the uric acid and activity of XO. However, the protein expression of XO was not altered (Appendix A).

## 4. Discussion

In this trial, we demonstrated that resveratrol had no significant effect on lipid profile but reduced serum uric acid and XO activity in a dose-response manner in subjects with dyslipidemia. A significant decrease of serum uric acid at eight weeks was observed in the 300 mg/d and 600 mg/d resveratrol groups but not in the 100 mg/d group, indicating resveratrol supplementation of 300 mg/d or more could produce a uric acid-lowering effect in subjects with dyslipidemia.

Current evidence suggests that hyperlipidemia is closely related to hyperuricemia. Indeed, it is well-established that hyperlipidemia and hyperuricemia are independent risk factors for CVD. Several epidemiological studies have revealed an independent relationship between elevated serum uric acid concentration and increased morbidity or mortality from atherosclerotic diseases [36]. Compared with healthy subjects, patients with hyperuricemia often display higher plasma triglyceride levels and atherogenic indexes [37]. In addition, serum uric acid is positively associated with triglyceride, TC, and LDL cholesterol and negatively associated with HDL cholesterol [38].

Resveratrol is a polyphenol whose beneficial effects on dyslipidemia and hyperuricemia have been demonstrated in animal models. However, clinical trials have yielded inconclusive findings on its effect on dyslipidemia, which may be attributed to the heterogeneity in drug doses across studies. For example, Simental et al. [39] demonstrated that 100 mg/d resveratrol for two months significantly reduced TC and triglyceride in dyslipidemia individuals. Chen et al. [35] reported that 300 mg/d resveratrol decreased LDL cholesterol and TC in subjects with nonalcoholic fatty liver disease (NAFLD) at 12 weeks. However, a study including 54 aged men showed that resveratrol supplementation (250 mg/d for 8 weeks) did not alter the lipid profile [40]. Likewise, another randomized controlled crossover study reported no change in HDL cholesterol, LDL cholesterol, or TC but a slight increase in triglyceride with 150 mg/d resveratrol for 4 weeks in 45 overweight and mildly obese subjects [41]. Another trial using a higher dose of resveratrol (1000 mg/d) for 3 months conducted by Mansour et al. [42] did not report any difference in lipid parameters in women with polycystic ovary syndrome. Similarly, C Apostolidou et al. [43] found no improvement in lipid levels with 1500 mg/d resveratrol for 4 weeks in patients with hypercholesterolemia. The present trial consistently showed that the doses of resveratrol from 100 mg/d to 600 mg/d yielded no significant benefit on the lipid profile. Therefore, the antilipidemic effect of resveratrol should be questioned with more caution.

Regarding the effect of resveratrol on hyperuricemia, animal studies showed that resveratrol reduced serum uric acid concentration by inhibiting XO to reduce uric acid production [44] and regulating organic ion transporters to increase uric acid excretion [45,46], which are inconsistent with the results of clinical studies. It has been reported that resveratrol dimers significantly reduced uric acid levels in healthy subjects [47,48]. As for resveratrol, there were neither clinical data documenting the effect on uric acid in patients with dyslipidemia nor evidence in diabetic patients, healthy smokers, or healthy subjects. In a trial of 192 patients with type 2 diabetes mellitus by Bo et al. [49], resveratrol (40 and 500 mg/d) intervention for 6 months had no significant impact on weight, BMI, blood pressure, glucose, glycated hemoglobin, insulin, uric acid, or interleukin-6 (IL-6). In another double-blind crossover trial conducted by the authors in 50 healthy adult smokers [50], resveratrol (500 mg/d) intervention lasting for 30 days significantly decreased C-reactive protein and triglyceride and increased total antioxidant status values, but without measurable effect on weight, blood pressure, glucose, insulin, uric acid, or TC. Similarly, resveratrol (100 mg/d) intervention for 90 days in Brazilian military firefighters resulted in significantly reduced inflammatory biomarkers IL-6 and tumor necrosis factor-α, but no effect was found on blood lipids, glucose, or uric acid [51].

Overall, the discrepancies reported above can be attributed to the heterogeneity in doses of resveratrol, intervention duration, and study subjects. Besides, no study has compared the effect of resveratrol with multiple doses. The present study provided hitherto undocumented evidence that 300 mg/d and 600 mg/d of resveratrol could reduce serum uric acid concentration in subjects with dyslipidemia, but no significant change was observed with 100 mg/d resveratrol.

In terms of the mechanism, many in vitro enzyme inhibition experiments have confirmed that resveratrol could significantly inhibit XO activity [52], but the effect has not been validated in any clinical trial. In the present trial, we provided compelling evidence that resveratrol could reduce serum XO activity in dyslipidemia in a dose-response manner, leading to corresponding decreases in uric acid. We further explored the mechanism of resveratrol on XO in human HepG2 cells and found that FFA treatment increased uric acid levels in the supernatants and intracellular XO activity, and resveratrol (10 μmol/L) significantly decreased the uric acid and activity of XO. Taken together, these results suggest that the uric acid-lowering effect of resveratrol can be attributed to a certain extent to decreased uric acid production mediated by decreasing XO activity, consistent with the literature [44,53]. Although the dosage of resveratrol in cell experiment (10 μmol/L) was higher than the physiological concentration reported [54], it is at a low dose level in previous in vitro studies [55,56]. Moreover, given that a long intervention duration was required for human research, the results of cell experiments can be used to support the findings of a clinical trial. In this respect, although numerous animal studies suggest that resveratrol may decrease serum uric acid by promoting uric acid excretion, it remains unclear whether resveratrol promotes uric acid excretion since 24 h urine volume was not collected from the subjects for measuring 24 h urine uric acid in our study, emphasizing the need for further research.

In addition, uric acid is widely acknowledged as an oxypurine with antioxidant properties and a major endogenous antioxidant in the human body. Resveratrol, as a natural exogenous antioxidant, may decrease the production of uric acid by reducing oxidative stress. In this regard, Xiao et al. [44] showed that resveratrol intervention in hyperuricemia rats reduced serum uric acid and XO and levels of SOD, a traditional oxidative stress biomarker, indicating that oxidative stress was alleviated. In clinical studies, Ali Movahed et al. [57] demonstrated that 500 mg/d resveratrol for 60 days decreased oxidative stress biomarker MDA levels and increased total antioxidant capacity in subjects with type 1 diabetes mellitus. However, in another NAFLD study, no changes in plasma lipids or oxidative stress biomarkers were observed after 3000 mg/d resveratrol was administered for 8 weeks [58]. Similarly, in the present study, oxidative stress biomarkers such as serum SOD, MDA, allantoin, GST, GSH, GSH-Px, urine 8-iso-PGF2α, and 8-OHdG were quantified, but no significant differences were found. Overall, there is no direct evidence that resveratrol can ameliorate oxidative stress in dyslipidemia, and antioxidant stress cannot account for its uric acid-lowering effect.

Importantly, the randomized, double-blinded, placebo-controlled trial design of this study minimized potential biases and maximized the homogeneity of subjects and comparability among groups. Moreover, precautions were taken to ensure study rigor throughout the data collection and analytical phases. Nevertheless, the most significant limitation of our study is that it was not designed or conducted to distinguish hyperlipidemia patients with hyperuricemia; instead, uric acid and XO activity were used as secondary outcomes. Therefore, it is unclear whether the dose and duration of resveratrol in the present trial are optimal for patients with hyperuricemia alone or with other metabolic diseases. Furthermore, we failed to detect resveratrol or its metabolites in subjects’ blood samples to assess their adherence since the subjects had consumed capsules ≥ 20 h before blood sample collection. In this respect, resveratrol reportedly has a plasma half-life of only 9.2 h [59].

## 5. Conclusions

Our study demonstrated that resveratrol supplementation for 8 weeks yielded no benefit on the lipid profile but decreased serum uric acid levels by inhibiting XO activity in a dose-response manner in subjects with dyslipidemia.

## Figures and Tables

**Figure 1 nutrients-15-00492-f001:**
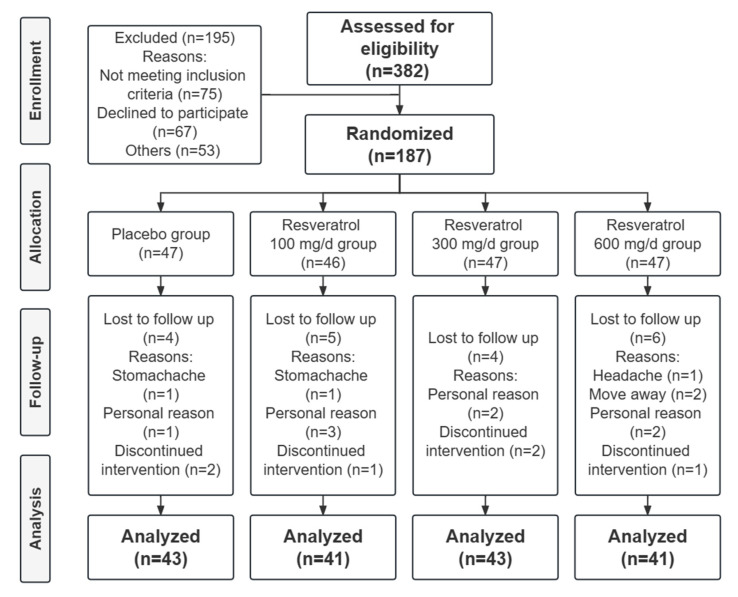
Flow diagram of the trial.

**Figure 2 nutrients-15-00492-f002:**
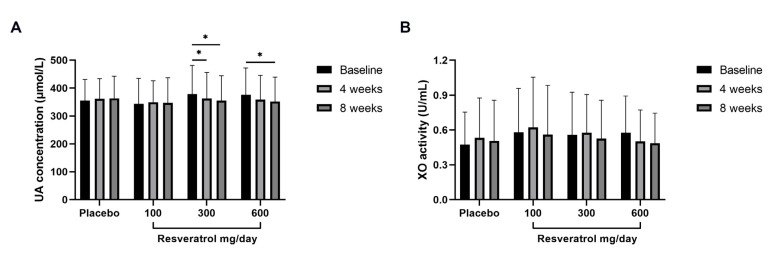
Absolute changes in uric acid concentration and XO activity within groups. Data are expressed as the mean ± SD measurements for each group. * *p* < 0.05. UA—uric acid; XO—xanthine oxidase.

**Figure 3 nutrients-15-00492-f003:**
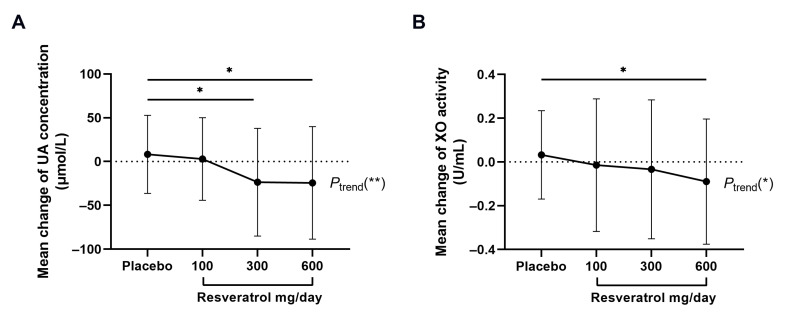
Relative changes in uric acid concentration and XO activity between groups. Data are expressed as the mean ± SD measurements for each group. * *p* < 0.05 and ** *p* < 0.01 between resveratrol groups and placebo group as well as linear trend analysis. UA—uric acid; XO—xanthine oxidase.

**Figure 4 nutrients-15-00492-f004:**
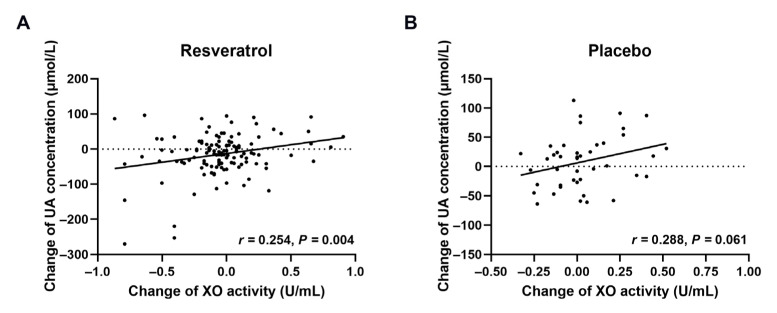
Correlation between changes in uric acid concentration and XO activity. Pearson’s correlation coefficients are noted for each plot. (**A**,**B**) represent the resveratrol groups (*n* = 125) and placebo group (*n* = 43), respectively. UA—uric acid; XO—xanthine oxidase.

**Figure 5 nutrients-15-00492-f005:**
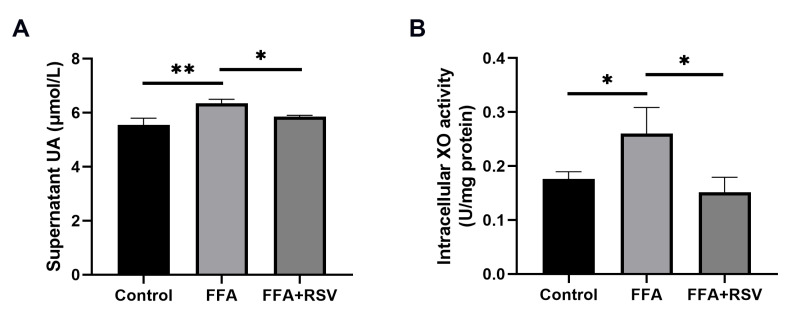
Effect of resveratrol on uric acid levels in the supernatants of HepG2 cells and intracellular XO activity. Data are expressed as the mean ± SD from the triplicate wells for 3 experiments. (**A**,**B**) represent the effect of resveratrol (10 μmol/L) on uric acid levels (**A**) and intracellular XO activity (**B**). * *p* < 0.05; ** *p* < 0.01. FFA—free fatty acids; RSV—resveratrol; UA—uric acid; XO—xanthine oxidase.

**Table 1 nutrients-15-00492-t001:** Baseline characteristics ^a^.

	Placebo (*n* = 43)	100 mg/d RSV (*n* = 41)	300 mg/d RSV (*n* = 43)	600 mg/d RSV (*n* = 41)	*p* ^b^
Age, years	61.30 ± 8.96	59.49 ± 8.70	61.14 ± 9.19	60.80 ± 8.32	0.947
Sex, M/F, *n*	13/30	11/30	14/29	13/28	0.945
Weight, kg	60.13 ± 9.25	61.21 ± 8.32	59.48 ± 7.22	61.69 ± 9.21	0.627
Current smoking, *n* (%)	2 (4.7)	2 (4.9)	1 (2.3)	2 (4.9)	0.920
Physical activities, MET-min/week	3363.34 ± 2292.33	3675.63 ± 2335.91	3530.37 ± 3327.86	3521.72 ± 2436.81	0.961
BMI, kg/m^2^	22.81 ± 4.33	23.72 ± 2.55	23.04 ± 1.95	24.16 ± 3.04	0.174
WC, cm	83.53 ± 8.34	85.18 ± 8.12	83.74 ± 6.83	86.22 ± 8.70	0.373
HC, cm	95.60 ± 4.95	96.72 ± 6.49	95.62 ± 4.51	96.98 ± 6.59	0.567
WHR, %	0.87 ± 0.07	0.88 ± 0.05	0.88 ± 0.05	0.89 ± 0.06	0.647
SBP, mmHg	119.07 ± 15.74	122.02 ± 18.19	124.93 ± 17.55	124.99 ± 15.58	0.313
DBP, mmHg	74.11 ± 9.84	74.24 ± 11.38	74.78 ± 9.38	76.83 ± 9.56	0.585
HR, bpm	78.41 ± 9.98	77.12 ± 11.32	75.86 ± 10.24	79.17 ± 9.79	0.474
TC, mmol/L	6.23 ± 0.87	6.28 ± 0.98	6.46 ± 1.08	6.29 ± 0.99	0.716
LDL cholesterol, mmol/L	4.04 ± 0.87	4.05 ± 1.00	4.25 ± 1.12	4.03 ± 0.95	0.692
HDL cholesterol, mmol/L	1.50 ± 0.41	1.38 ± 0.35	1.41 ± 0.43	1.39 ± 0.43	0.498
Triglyceride, mmol/L	1.75 ± 0.94	2.20 ± 1.08	2.01 ± 1.15	2.27 ± 1.33	0.159
ApoA1, g/L	1.64 ± 0.29	1.56 ± 0.28	1.58 ± 0.30	1.57 ± 0.29	0.621
ApoB, g/L	1.26 ± 0.21	1.31 ± 0.25	1.38 ± 0.27	1.31 ± 0.20	0.125
ApoA1/apoB	1.33 ± 0.33	1.26 ± 0.41	1.18 ± 0.29	1.22 ± 0.26	0.219
Uric acid, μmol/L	355.09 ± 75.99	343.68 ± 91.58	378.37 ± 102.62	376.37 ± 95.94	0.246
Glucose, mmol/L	5.57 ± 0.90	5.73 ± 1.20	5.72 ± 0.91	5.70 ± 0.85	0.846
Insulin, μU/mL	10.70 ± 6.15	12.63 ± 10.90	10.13 ± 6.07	11.40 ± 6.37	0.474
HOMA-IR	2.73 ± 1.82	3.47 ± 4.04	2.63 ± 1.84	2.95 ± 1.96	0.452
SOD, U/mL	163.39 ± 15.26	164.28 ± 13.35	158.97 ± 13.05	165.82 ± 17.07	0.174
MDA, ng/mL	5.94 ± 2.15	5.82 ± 1.75	5.71 ± 1.85	6.08 ± 1.83	0.831
Allantoin, μmol/L	24.24 ± 15.93	22.23 ± 11.59	23.94 ± 15.37	26.04 ± 17.05	0.726
GST, ng/mL	0.65 ± 0.24	0.67 ± 0.26	0.61 ± 0.22	0.63 ± 0.22	0.736
GSH, μg/mL	50.26 ± 23.16	48.05 ± 18.92	52.11 ± 19.98	50.99 ± 21.72	0.842
GSH-Px, U/mL	192.56 ± 120.16	176.14 ± 67.16	190.67 ± 72.48	187.31 ± 80.59	0.832
Urine 8-iso-PGF2α, pg/mg creatinine	0.40 ± 0.35	0.38 ± 0.31	0.35 ± 0.29	0.38 ± 0.29	0.890
Urine 8-OHdG, pg/mg creatinine	92.07 ± 88.36	82.44 ± 58.75	80.45 ± 75.10	78.64 ± 55.93	0.824

^a^ The results are presented as mean ± SD for continuous variables and *n* (%) for categorical variables. RSV—resveratrol group; M—male; F—female; WC—waist circumference; HC—hip circumference; WHR—waist-hip ratio; SBP—systolic blood pressure; DBP—diastolic blood pressure; HR—heart rate; TC—total cholesterol; SOD—superoxide dismutase; MDA—malonaldehyde; GST—glutathione S-transferase; GSH—glutathione; GSH-Px—glutathione peroxidase; 8-iso-PGF2α—8-iso-prostaglandin F2α; 8-OHdG—8-hydroxy-2′-deoxyguanosine. ^b^
*p* values are for comparison among the four groups (Either one-way ANOVA or chi-square test for independent data).

**Table 2 nutrients-15-00492-t002:** Daily dietary intakes of total energy and nutrients at baseline and after 8 weeks of treatment ^a^.

	Placebo (*n* = 43)	100 mg/d RSV (*n* = 41)	300 mg/d RSV (*n* = 43)	600 mg/d RSV (*n* = 41)	*p* ^b^
Total energy, kcal/day					
Baseline	1707.31 ± 426.51	1664.53 ± 349.31	1608.03 ± 385.70	1688.25 ± 519.33	0.724
8 weeks	1694.26 ± 409.49	1745.70 ± 390.77	1602.58 ± 384.07	1696.12 ± 419.87	0.425
Protein, g/day					
Baseline	77.87 ± 28.69	68.09 ± 25.46	66.82 ± 20.29	72.58 ± 28.71	0.195
8 weeks	77.74 ± 26.11	76.60 ± 21.19	69.04 ± 29.65	73.14 ± 24.64	0.396
Fat, g/day					
Baseline	75.14 ± 23.37	69.10 ± 25.00	64.28 ± 20.75	67.51 ± 20.87	0.159
8 weeks	72.98 ± 21.26	75.38 ± 22.33	65.54 ± 19.81	74.16 ± 19.13	0.124
Carbohydrate, g/day					
Baseline	179.16 ± 56.29	198.48 ± 100.37	190.73 ± 68.89	210.64 ± 106.46	0.388
8 weeks	187.52 ± 58.83	205.04 ± 84.67	193.21 ± 66.95	198.30 ± 67.68	0.700
Dietary fiber, g/day					
Baseline	13.43 ± 8.07	10.66 ± 6.75	12.02 ± 8.48	12.65 ± 7.32	0.410
8 weeks	14.44 ± 10.62	12.22 ± 6.80	11.75 ± 11.22	11.96 ± 5.73	0.481
Cholesterol, mg/day					
Baseline	454.50 ± 196.67	439.27 ± 233.49	404.69 ± 206.33	452.78 ± 215.25	0.681
8 weeks	516.51 ± 211.21	502.36 ± 192.82	428.11 ± 211.86	461.86 ± 186.21	0.170
Vitamin A, μg retinol equivalent/day					
Baseline	752.88 ± 343.70	712.75 ± 485.53	658.63 ± 356.38	692.70 ± 350.01	0.722
8 weeks	779.78 ± 428.21	749.09 ± 354.98	686.08 ± 358.88	788.16 ± 511.28	0.667
Vitamin E, mg/day					
Baseline	23.61 ± 9.83	20.00 ± 6.04	20.92 ± 8.03	22.71 ± 7.28	0.149
8 weeks	23.46 ± 12.88	22.39 ± 9.22	21.04 ± 8.62	24.00 ± 8.84	0.546
Vitamin C, mg/day					
Baseline	101.93 ± 63.72	102.51 ± 50.53	105.31 ± 50.39	101.54 ± 53.61	0.989
8 weeks	105.55 ± 79.54	108.31 ± 51.23	102.75 ± 49.95	103.31 ± 55.10	0.974
Resveratrol, mg/day					
Baseline	11.48 ± 8.75	10.55 ± 7.00	10.76 ± 5.27	10.53 ± 6.07	0.912
8 weeks	11.61 ± 9.16	11.13 ± 6.19	10.39 ± 6.59	11.81 ± 8.32	0.834

^a^ Data are presented as mean ± SD. RSV, resveratrol group. ^b^ *p* values are for comparison among the four groups at baseline and after 8 weeks of intervention (One-way ANOVA for independent data).

**Table 3 nutrients-15-00492-t003:** Effect of resveratrol on lipid profile ^a^.

	Placebo (*n* = 43)	100 mg/d RSV (*n* = 41)	300 mg/d RSV (*n* = 43)	600 mg/d RSV (*n* = 41)	*p* ^b^	*p* Trend
TC, mmol/L						
Baseline	6.23 ± 0.87	6.28 ± 0.98	6.46 ± 1.08	6.29 ± 0.99	0.716	
4 weeks	6.14 ± 0.93	6.27 ± 0.89	6.43 ± 1.19	6.44 ± 1.14		
8 weeks	6.16 ± 1.02	6.34 ± 0.98	6.52 ± 1.25	6.43 ± 1.30		
4 weeks change	−0.09 ± 0.56	−0.01 ± 0.50	−0.03 ± 0.79	0.15 ± 0.74	0.392	0.125
8 weeks change	−0.07 ± 0.67	0.06 ± 0.85	0.06 ± 0.86	0.15 ± 0.69	0.636	0.224
LDL cholesterol, mmol/L						
Baseline	4.04 ± 0.87	4.05 ± 1.00	4.25 ± 1.12	4.03 ± 0.95	0.692	
4 weeks	4.04 ± 0.93	3.98 ± 0.95	4.20 ± 1.22	4.28 ± 1.16		
8 weeks	3.98 ± 0.94	4.02 ± 0.97	4.27 ± 1.27	4.24 ± 1.25		
4 weeks change	0.00 ± 0.54	−0.08 ± 0.56	−0.05 ± 0.77	0.26 ± 0.74	0.091	0.081
8 weeks change	−0.06 ± 0.64	−0.03 ± 0.85	0.02 ± 0.88	0.22 ± 0.74	0.380	0.107
HDL cholesterol, mmol/L						
Baseline	1.50 ± 0.41	1.38 ± 0.35	1.41 ± 0.43	1.39 ± 0.43	0.498	
4 weeks	1.52 ± 0.42	1.33 ± 0.32	1.37 ± 0.40	1.35 ± 0.38		
8 weeks	1.52 ± 0.43	1.37 ± 0.35	1.40 ± 0.40	1.36 ± 0.39		
4 weeks change	0.02 ± 0.19	−0.05 ± 0.19	−0.04 ± 0.13	−0.04 ± 0.12	0.189	0.150
8 weeks change	0.02 ± 0.17	−0.01 ± 0.19	−0.01 ± 0.15	−0.03 ± 0.16	0.586	0.193
Triglyceride, mmol/L						
Baseline	1.75 ± 0.94	2.20 ± 1.08	2.01 ± 1.15	2.27 ± 1.33	0.159	
4 weeks	1.61 ± 0.88	2.41 ± 1.41	2.21 ± 1.31	2.32 ± 1.50		
8 weeks	1.77 ± 1.06	2.32 ± 1.31	2.22 ± 1.20	2.30 ± 1.58		
4 weeks change	−0.14 ± 0.59	0.21 ± 0.96	0.20 ± 0.68	0.05 ± 0.55	0.084	0.731
8 weeks change	0.02 ± 0.86	0.12 ± 0.91	0.21 ± 0.70	0.03 ± 0.85	0.719	0.848
ApoA1, g/L						
Baseline	1.64 ± 0.29	1.56 ± 0.28	1.58 ± 0.30	1.57 ± 0.29	0.621	
4 weeks	1.60 ± 0.26	1.52 ± 0.26	1.56 ± 0.28	1.53 ± 0.26		
8 weeks	1.61 ± 0.26	1.56 ± 0.29	1.59 ± 0.27	1.54 ± 0.26		
4 weeks change	−0.04 ± 0.19	−0.04 ± 0.13	−0.03 ± 0.11	−0.04 ± 0.10	0.937	0.821
8 weeks change	−0.03 ± 0.20	0.00 ± 0.14	0.00 ± 0.12	−0.04 ± 0.11	0.440	0.897
ApoB, g/L						
Baseline	1.26 ± 0.21	1.31 ± 0.25	1.38 ± 0.27	1.31 ± 0.20	0.125	
4 weeks	1.24 ± 0.22	1.30 ± 0.22	1.37 ± 0.30	1.37 ± 0.26		
8 weeks	1.25 ± 0.23	1.31 ± 0.23	1.41 ± 0.33	1.38 ± 0.28		
4 weeks change	− 0.02 ± 0.14	−0.01 ± 0.14	−0.01 ± 0.21	0.06 ± 0.18	0.106	0.043
8 weeks change	0.00 ± 0.14	0.00 ± 0.20	0.03 ± 0.25	0.07 ± 0.16	0.278	0.062
ApoA1/apoB						
Baseline	1.33 ± 0.33	1.26 ± 0.41	1.18 ± 0.29	1.22 ± 0.26	0.219	
4 weeks	1.32 ± 0.32	1.22 ± 0.32	1.18 ± 0.32	1.14 ± 0.25		
8 weeks	1.32 ± 0.33	1.24 ± 0.36	1.19 ± 0.35	1.15 ± 0.24		
4 weeks change	0.00 ± 0.16	−0.03 ± 0.21	0.00 ± 0.20	−0.08 ± 0.17	0.165	0.130
8 weeks change	0.00 ± 0.17	−0.01 ± 0.27	0.01 ± 0.25	−0.08 ± 0.15	0.256	0.164

^a^ Data are presented as mean ± SD. RSV, resveratrol group; TC—total cholesterol. ^b^ *p* values are for comparison among the four groups (One-way ANOVA for independent data).

**Table 4 nutrients-15-00492-t004:** Effect of resveratrol on uric acid and XO activity ^a^.

	Placebo (*n* = 43)	100 mg/d RSV (*n* = 41)	300 mg/d RSV (*n* = 43)	600 mg/d RSV (*n* = 41)	*p*	*p* Trend
Uric acid, μmol/L						
Baseline	355.09 ± 75.99	343.68 ± 91.58	378.37 ± 102.62	376.37 ± 95.94	0.246	
4 weeks	361.69 ± 72.29	349.24 ± 77.27	362.51 ± 93.51 ^b^	358.44 ± 87.31		
8 weeks	363.28 ± 79.49	346.63 ± 90.58	354.77 ± 89.00 ^b^	352.00 ± 87.52 ^b^		
4 weeks change	6.59 ± 43.45	5.56 ± 76.15	−15.86 ± 44.40	−17.93 ± 63.42	0.091	0.019
8 weeks change	8.19 ± 44.60	2.95 ± 47.16	−23.60 ± 61.53 ^c^	−24.37 ± 64.24 ^c^	0.008	0.001
XO activity, U/mL						
Baseline	0.47 ± 0.28	0.58 ± 0.38	0.56 ± 0.37	0.58 ± 0.32	0.436	
4 weeks	0.53 ± 0.34	0.62 ± 0.43	0.58 ± 0.33	0.50 ± 0.27		
8 weeks	0.51 ± 0.35	0.56 ± 0.42	0.53 ± 0.33	0.49 ± 0.26		
4 weeks change	0.06 ± 0.22	0.04 ± 0.32	0.02 ± 0.34	−0.07 ± 0.32	0.195	0.045
8 weeks change	0.03 ± 0.20	−0.01 ± 0.30	−0.03 ± 0.32	−0.09 ± 0.29 ^d^	0.258	0.048

^a^ Data are presented as mean ± SD. RSV—resveratrol group; XO—xanthine oxidase. ^b^ Compared within the group from baseline to follow-up (Student’s *t*-test for paired data), *p* < 0.05 indicated a significant difference. ^c^ Compared between the placebo group and resveratrol groups (One-way ANOVA for independent data), *p* < 0.05 indicated a significant difference. ^d^ Compared between the placebo group and resveratrol group (Student’s *t*-test for independent samples), *p* < 0.05 indicated a significant difference.

**Table 5 nutrients-15-00492-t005:** Effect of resveratrol on glucose and insulin ^a^.

	Placebo (*n* = 43)	100 mg/d RSV (*n* = 41)	300 mg/d RSV (*n* = 43)	600 mg/d RSV (*n* = 41)	*p* ^b^	*p* Trend
Glucose, mmol/L						
Baseline	5.57 ± 0.90	5.73 ± 1.20	5.72 ± 0.91	5.70 ± 0.85	0.846	
4 weeks	5.54 ± 0.70	5.80 ± 1.08	5.77 ± 0.79	5.77 ± 0.98		
8 weeks	5.62 ± 0.73	5.64 ± 1.14	5.76 ± 0.79	5.88 ± 1.04		
4 weeks change	−0.03 ± 0.50	0.06 ± 0.57	0.05 ± 0.39	0.08 ± 0.65	0.808	0.424
8 weeks change	0.05 ± 0.35	−0.10 ± 0.61	0.03 ± 0.54	0.19 ± 0.64	0.138	0.160
Insulin, μU/mL						
Baseline	10.70 ± 6.15	12.63 ± 10.90	10.13 ± 6.07	11.40 ± 6.37	0.474	
4 weeks	10.04 ± 5.61	11.75 ± 7.25	10.36 ± 6.62	11.50 ± 6.63		
8 weeks	9.86 ± 6.73	11.25 ± 6.52	10.40 ± 5.95	12.45 ± 8.68		
4 weeks change	−0.66 ± 3.87	−0.88 ± 6.26	0.24 ± 4.05	0.10 ± 4.00	0.616	0.288
8 weeks change	−0.84 ± 4.88	−1.38 ± 9.92	0.27 ± 4.02	1.05 ± 8.30	0.409	0.139
HOMA-IR						
Baseline	2.73 ± 1.82	3.47 ± 4.04	2.63 ± 1.84	2.95 ± 1.96	0.452	
4 weeks	2.53 ± 1.52	3.13 ± 2.33	2.71 ± 1.84	3.03 ± 2.00		
8 weeks	2.51 ± 1.78	2.92 ± 1.97	2.71 ± 1.66	3.38 ± 2.90		
4 weeks change	−0.20 ± 1.10	−0.34 ± 2.27	0.08 ± 1.22	0.08 ± 1.38	0.523	0.241
8 weeks change	−0.22 ± 1.28	−0.55 ± 3.74	0.08 ± 1.34	0.43 ± 2.71	0.315	0.131

^a^ Data are presented as mean ± SD. RSV—resveratrol group. ^b^ *p* values are for comparison among the four groups (One-way ANOVA for independent data).

**Table 6 nutrients-15-00492-t006:** Effect of resveratrol on oxidative stress biomarkers ^a^.

	Placebo (*n* = 43)	100 mg/d RSV (*n* = 41)	300 mg/d RSV (*n* = 43)	600 mg/d RSV (*n* = 41)	*p* ^b^	*p* Trend
SOD, U/mL						
Baseline	163.39 ± 15.26	164.28 ± 13.35	158.97 ± 13.05	165.82 ± 17.07	0.174	
4 weeks	158.21 ± 12.80 ^c^	157.77 ± 17.24 ^c^	157.06 ± 12.77	158.72 ± 13.92 ^c^		
8 weeks	160.92 ± 12.56	162.22 ± 16.22	159.49 ± 13.13	162.92 ± 13.25		
4 weeks change	−5.18 ± 11.22	−6.51 ± 18.82	−1.91 ± 12.48	−7.09 ± 11.87	0.320	0.905
8 weeks change	−2.47 ± 13.49	−2.06 ± 19.63	0.53 ± 12.08	−2.90 ± 10.52	0.688	0.896
MDA, ng/mL						
Baseline	5.94 ± 2.15	5.82 ± 1.75	5.71 ± 1.85	6.08 ± 1.83	0.831	
4 weeks	4.05 ± 1.91	4.00 ± 0.81	3.64 ± 0.94	3.80 ± 0.97		
8 weeks	3.61 ± 1.33	3.71 ± 1.41	3.67 ± 1.25	3.40 ± 1.11		
4 weeks change	−1.89 ± 2.88	−1.82 ± 1.89	−2.07 ± 1.83	−2.27 ± 1.91	0.783	0.354
8 weeks change	−2.33 ± 2.31	−2.12 ± 2.09	−2.04 ± 2.44	−2.68 ± 2.36	0.585	0.542
Allantoin, μmol/L						
Baseline	24.24 ± 15.93	22.23 ± 11.59	23.94 ± 15.37	26.04 ± 17.05	0.726	
4 weeks	22.87 ± 12.90	20.82 ± 12.44	20.94 ± 12.00	26.20 ± 27.83		
8 weeks	20.73 ± 10.51	24.12 ± 17.24	22.16 ± 13.20	23.08 ± 19.54		
4 weeks change	−1.38 ± 17.71	−1.41 ± 15.78	−3.01 ± 18.69	0.16 ± 27.40	0.917	0.831
8 weeks change	−3.52 ± 15.50	1.90 ± 20.00	−1.79 ± 20.24	−2.96 ± 19.21	0.557	0.877
GST, ng/mL						
Baseline	0.65 ± 0.24	0.67 ± 0.26	0.61 ± 0.22	0.63 ± 0.22	0.736	
4 weeks	0.57 ± 0.23	0.61 ± 0.25	0.54 ± 0.22	0.58 ± 0.25		
8 weeks	0.58 ± 0.23	0.63 ± 0.25	0.55 ± 0.23	0.59 ± 0.24		
4 weeks change	−0.07 ± 0.11	−0.06 ± 0.12	−0.07 ± 0.12	−0.05 ± 0.19	0.871	0.654
8 weeks change	−0.07 ± 0.13	−0.04 ± 0.15	−0.06 ± 0.13	−0.04 ± 0.18	0.784	0.601
GSH, μg/mL						
Baseline	50.26 ± 23.16	48.05 ± 18.92	52.11 ± 19.98	50.99 ± 21.72	0.842	
4 weeks	57.69 ± 24.13	53.63 ± 21.65	60.31 ± 26.16	57.51 ± 23.75		
8 weeks	56.66 ± 24.93	50.53 ± 22.13	59.45 ± 26.37	55.37 ± 24.03		
4 weeks change	7.43 ± 14.25	5.57 ± 11.62	8.20 ± 14.68	6.52 ± 17.17	0.858	0.992
8 weeks change	6.40 ± 14.55	2.48 ± 12.37	7.34 ± 14.75	4.38 ± 15.17	0.411	0.904
GSH-Px, U/mL						
Baseline	192.56 ± 120.16	176.14 ± 67.16	190.67 ± 72.48	187.31 ± 80.59	0.832	
4 weeks	212.38 ± 89.94	196.45 ± 78.67	236.16 ± 131.33	212.00 ± 90.24		
8 weeks	208.06 ± 102.03	190.10 ± 84.22	207.52 ± 85.12	203.77 ± 88.29		
4 weeks change	19.82 ± 89.52	20.31 ± 43.82	45.49 ± 82.45	24.69 ± 66.48	0.319	0.431
8 weeks change	15.51 ± 57.16	13.96 ± 52.13	16.85 ± 67.68	16.46 ± 57.34	0.996	0.888
Urine 8-iso-PGF2α, pg/mg creatinine						
Baseline	0.40 ± 0.35	0.38 ± 0.31	0.35 ± 0.29	0.38 ± 0.29	0.890	
4 weeks	0.30 ± 0.21	0.23 ± 0.16	0.28 ± 0.26	0.25 ± 0.19		
8 weeks	0.24 ± 0.13	0.24 ± 0.15	0.25 ± 0.23	0.22 ± 0.17		
4 weeks change	−0.11 ± 0.34	−0.14 ± 0.32	−0.07 ± 0.29	−0.12 ± 0.31	0.720	0.915
8 weeks change	−0.16 ± 0.32	−0.14 ± 0.30	−0.10 ± 0.33	−0.16 ± 0.27	0.819	0.873
Urine 8-OHdG, pg/mg creatinine						
Baseline	92.07 ± 88.36	82.44 ± 58.75	80.45 ± 75.10	78.64 ± 55.93	0.824	
4 weeks	62.68 ± 33.14	52.52 ± 27.75	63.17 ± 44.95	58.21 ± 31.04		
8 weeks	51.39 ± 22.45	52.64 ± 30.26	53.16 ± 36.32	48.71 ± 24.93		
4 weeks change	−29.40 ± 78.14	−29.93 ± 54.98	−17.28 ± 69.38	−20.43 ± 61.23	0.763	0.392
8 weeks change	−40.68 ± 77.93	−29.81 ± 52.87	−27.29 ± 78.45	−29.94 ± 54.24	0.799	0.455

^a^ Data are presented as mean ± SD. RSV—resveratrol group; SOD—superoxide dismutase; MDA—malonaldehyde; GST—glutathione S-transferase; GSH—glutathione; GSH-Px—glutathione peroxidase; 8-iso-PGF2α—8-iso-prostaglandin F2α; 8-OHdG—8-hydroxy-2′-deoxyguanosine. ^b^
*p* values are for comparison among the four groups (One-way ANOVA for independent data). ^c^ Compared within the group from baseline to follow-up (Student’s *t*-test for paired data), *p* < 0.05 indicated a significant difference.

## Data Availability

The data presented in this study are available on request from the corresponding author.

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
