# Peer review of "A Randomized Trial on Resveratrol Supplement Affecting Lipid Profile and Other Metabolic Markers in Subjects with Dyslipidemia"

_nutrients, 2023, doi:10.3390/nu15030492_

Round 1

Reviewer 1 Report

This study provides an interesting information. However, it is necessary to improve this manuscript. 

1.      Why did this cell culture study examine the effect of a single dose of resveratrol in HepG2 cells? Readers may doubt the authors did show only a good data with a single dose of resveratrol in the cell culture experiment.

2.      How about the possible underlying mechanisms of the effect of resveratrol on XO expression in HepG2 cells?

3.      The data of Table 4 appear to show that in the case of higher uric acid level, the effect of resveratrol might more effective. How do you consider?

4.      This study shows the statistically significant reduction in serum uric acid by the treatment of resveratrol at 600mg/d. However, the degree of the reduction was very slight (4~6% reduction)? Is the small change meaningful as a matter of clinical fact? 

Author Response

Point 1: Why did this cell culture study examine the effect of a single dose of resveratrol in HepG2 cells? Readers may doubt the authors did show only a good data with a single dose of resveratrol in the cell culture experiment.

Response 1: We gratefully thank the Reviewer for this comment. Our cell culture experiments were designed to confirm the findings from human intervention. The physiological concentrations of resveratrol in the subjects' plasma were too low to be detected, making it difficult to select different resveratrol doses for the cell culture experiment. Therefore, the appropriate dose used in cell experiments in our study was based on the previous literature. In this respect, we chose a low dose of resveratrol in cell experiments, as reported in previous studies. Although it was higher than the physiological concentration of resveratrol, considering that the intervention time of the cell experiment was 48h, significantly shorter than the duration in the clinical study, the results of the cell experiment were consistent with the results of clinical research to a certain extent.

Point 2: How about the possible underlying mechanisms of the effect of resveratrol on XO expression in HepG2 cells?

Response 2: Although it is well-established that resveratrol inhibits intracellular XO activity, it remains unclear whether resveratrol can reduce the expression of XO. PI3K/AKT is a well-known upstream regulatory pathway of XO. Interestingly, Zhang et al.[1] showed that liraglutide inhibited XO expression through the PI3K/Akt/SR-Ca(2+) pathway and protected endothelial cells against hypoxia/reoxygenation injury. Huang et al.[2] found that anthocyanins decreased XO levels in human umbilical vein endothelial cells through the PI3K/Akt signaling pathway and improved high-glucose-induced cell damage. Resveratrol has been reported to activate the PI3K/AKT pathway in different pathological models. In this regard, resveratrol could attenuate oxidative stress-induced intestinal barrier injury and inhibit paclitaxel-induced neuropathic pain via the PI3K/Akt signaling pathway[3, 4]. Based on the above reports, we speculate that resveratrol may reduce XO expression in HepG2 cells through PI3K/AKT pathway. However, our cell experiment yielded no positive findings, emphasizing the need for further research after improving the experimental protocol.

Point 3: The data of Table 4 appear to show that in the case of higher uric acid level, the effect of resveratrol might more effective. How do you consider?

Response 3: We are grateful to the Reviewer for this insightful question, which prompted us to recheck the statistical analysis of the data. First, a one-way ANOVA analysis of each group's baseline uric acid values showed no significant difference, suggesting that the baseline uric acid was comparable among the groups. Subsequently, we conducted a one-way ANCOVA analysis to exclude the influence of baseline values (not mentioned in the paper). The results showed that after adjusting the baseline value of uric acid as a covariable, the P value of the group variable remained less than 0.05, indicating that the mean difference of uric acid among the groups remained statistically significant. Since the one-way ANCOVA analysis was not carried out on other outcome variables, we did not include this part of statistics in the paper, considering the homogeneity of statistical analysis in the whole paper. Although the effects of resveratrol might be more effective at high uric acid levels, no significant difference was found. However, it is worth noting that two studies on the effect of plant compounds on lowering uric acid reported that their effects were more significant in subjects with higher uric acid levels [5, 6], consistent with animal studies. Therefore, although no statistical significance was observed for the effects of resveratrol, it is worth further research.

Point 4: This study shows the statistically significant reduction in serum uric acid by the treatment of resveratrol at 600mg/d. However, the degree of the reduction was very slight (4~6% reduction)? Is the small change meaningful as a matter of clinical fact?

Response 4: The effect of plant compound resveratrol on lowering uric acid (4~6% reduction) was not comparable to traditional uric acid-lowering drugs. For example, 300 mg/d allopurinol for 12 weeks reduced uric acid by 39%[7]. However, compared with other plant compounds, resveratrol exerted a moderate effect on reducing uric acid levels. For example, 400 mg/d decaffeinated green tea polyphenols reduced uric acid by 12% after 12 weeks[8], 500 mg/d quercetin for 4 weeks resulted in an 8% reduction in uric acid[6], and 300 mg/d citrus bergamia and cynara cardunculus reduced uric acid by 1% for 6 weeks[5]. Therefore, resveratrol represents a promising approach to lower uric acid levels in individuals with supraoptimal blood uric acid levels but who have not yet developed any disease. In addition, as mentioned in the last paragraph of the discussion section, one of the drawbacks of this RCT is that uric acid was used only as a secondary outcome. To objectively verify the clinical effect of resveratrol on lowering uric acid, hyperuricemia rather than dyslipidemia should be included as the criterion.

Reference

  1. Zhang Y, Zhou H, Wu W, Shi C, Hu S, Yin T, Ma Q, Han T, Zhang Y, Tian F, et al: Liraglutide protects cardiac microvascular endothelial cells against hypoxia/reoxygenation injury through the suppression of the SR-Ca(2+)-XO-ROS axis via activation of the GLP-1R/PI3K/Akt/survivin pathways. Free Radic Biol Med 2016;95:278-292.
  2. Huang W, Hutabarat RP, Chai Z, Zheng T, Zhang W, Li D: Antioxidant Blueberry Anthocyanins Induce Vasodilation via PI3K/Akt Signaling Pathway in High-Glucose-Induced Human Umbilical Vein Endothelial Cells. Int J Mol Sci 2020;21(5).
  3. Li X, Yang S, Wang L, Liu P, Zhao S, Li H, Jiang Y, Guo Y, Wang X: Resveratrol inhibits paclitaxel-induced neuropathic pain by the activation of PI3K/Akt and SIRT1/PGC1خ▒ pathway. J Pain Res 2019;12:879-890.
  4. Zhuang Y, Wu H, Wang X, He J, He S, Yin Y: Resveratrol Attenuates Oxidative Stress-Induced Intestinal Barrier Injury through PI3K/Akt-Mediated Nrf2 Signaling Pathway. Oxid Med Cell Longev 2019;2019:7591840.
  5. Ferro Y, Maurotti S, Mazza E, Pujia R, Sciacqua A, Musolino V, Mollace V, Pujia A, Montalcini T: Citrus Bergamia and Cynara Cardunculus Reduce Serum Uric Acid in Individuals with Non-Alcoholic Fatty Liver Disease. Medicina (Kaunas) 2022;58(12).
  6. Shi Y, Williamson G: Quercetin lowers plasma uric acid in pre-hyperuricaemic males: a randomised, double-blinded, placebo-controlled, cross-over trial. Br J Nutr 2016;115(5):800-806.
  7. Jalal DI, Decker E, Perrenoud L, Nowak KL, Bispham N, Mehta T, Smits G, You Z, Seals D, Chonchol M, et al: Vascular Function and Uric Acid-Lowering in Stage 3 CKD. J Am Soc Nephrol 2017;28(3):943-952.
  8. Xie L, Tang Q, Yao D, Gu Q, Zheng H, Wang X, Yu Z, Shen X: Effect of Decaffeinated Green Tea Polyphenols on Body Fat and Precocious Puberty in Obese Girls: A Randomized Controlled Trial. Front Endocrinol (Lausanne) 2021;12:736724.

Reviewer 2 Report

This is a scientifically sound study, which demonstrates that the findings of animal studies do not always translate to humans. I think it is important to publish these null findings. In places, the English language is awkward (e.g., missing articles), so I recommended a copyedit by a native English language speaker.

Abstract
Line 18: Add an “a” before well-established, …with a well-established…
Line 20: Change “was to observe” to examined, …this study examined whether…
Line 28: Change “Besides” to Furthermore, …Furthermore, xanthine oxidase…

Graphical Abstract
The graphical abstract is very well done. However, I do not see how it communicates that resveratrol supplementation had no significant effect on lipid profile.

Introduction
Lines 48-49: Add a comma after hyperuricemia, change “are” to is, and add an a after it, make abnormalities singular care, …Hyperuricemia, or high serum uric acid concentration, is a common metabolic abnormality in patients with dyslipidemia...
Line 50: Add of individuals with between percentage and dysplipedima, ….It has been reported that 25.9% of individuals with dyslipidemia had hyperuricemia…
Line 84: Delete “and so on”
Line 86 Change “included” to were, …The inclusion criteria were…

Materials and Methods
The design has excellent scientific rigor and is well-described.
Lines 109-110: For the 24-hour recalls, was a specific process followed? Did a trained researcher interview the study participants?

Results
Line 175, be more explicit about how many individuals and in which groups stopped taking the supplement because of mild adverse reactions or personal reasons.
Figure 1, enrollment is spelled incorrectly.

Conclusions
The conclusion is supported by the results.

Author Response

Point 1: Abstract

Line 18: Add an “a” before well-established, …with a well-established…

Line 20: Change “was to observe” to examined, …this study examined whether…

Line 28: Change “Besides” to Furthermore, …Furthermore, xanthine oxidase…

Response 1: Thanks for your valuable comments. We are terribly sorry for our awkward English language and we have asked a native English language speaker to edit it. We have modified the Abstract part according to your suggestions.

Point 2: Graphical Abstract

The graphical abstract is very well done. However, I do not see how it communicates that resveratrol supplementation had no significant effect on lipid profile.

Response 2: Thank you for your compliment and suggestion. We have replaced the first small image at the top right to illustrate that resveratrol supplementation had no significant effect on lipid profile.

Point 3: Introduction

Lines 48-49: Add a comma after hyperuricemia, change “are” to is, and add an a after it, make abnormalities singular care, …Hyperuricemia, or high serum uric acid concentration, is a common metabolic abnormality in patients with dyslipidemia...

Line 50: Add of individuals with between percentage and dysplipedima, ….It has been reported that 25.9% of individuals with dyslipidemia had hyperuricemia…

Line 84: Delete “and so on”

Line 86 Change “included” to were, …The inclusion criteria were…

Response 3: Thanks for your thoughtful comments. We have modified the Introduction part according to your suggestions.

Point 4: Materials and Methods

Lines 109-110: For the 24-hour recalls, was a specific process followed? Did a trained researcher interview the study participants?

Response 4: Thanks for your suggestions. We have added some practical details about the 24-hour recalls as you suggested (revised version: page 4 line 113-114).

Point 5: Results

Line 175, be more explicit about how many individuals and in which groups stopped taking the supplement because of mild adverse reactions or personal reasons.

Figure 1, enrollment is spelled incorrectly.

Response 5: Thanks for your constructive comments. The details on the mild adverse reactions have been added (revised version: page 6 line 181-183). We are very sorry for the poor spelling error and we have corrected it.